# Generating the evidence to support the establishment of a Respiratory Syncytial Virus surveillance system in Cameroon: A study protocol

John Njuma Libwea [1,2,3,4]*, Linda Esso[1], Andreas Ateke Njoh[5,6], Che Henry Ngwa[4,7,8], Vivienne Ngomba Armelle[1], Chanceline Bilounga Ndongo[1], Georges Alain Etoundi Mballa[1], Bright I. Nwaru[4,11], Dan Weinberger[12], Richard Njouom[9,10], Sinata Koulla-Shiro[10]

1 Directorate for Disease Control, Epidemics and Pandemics Ministry of Public Health, Yaoundé, Cameroon, 2 Cameroon Academy of Sciences, Yaoundé, Cameroon, 3 Health Sciences Unit, Faculty of Social Sciences, Tampere University, Tampere, Finland, 4 African Science Frontiers Initiatives [ASFI], Lagos, Nigeria, 5 Central Coordination, Expanded Programme on Immunization, Yaoundé, Cameroon, 6 School of Global Health and Bioethics, Euclid University, Bangui, Central Africa Republic, 7 Department of Epidemiology and Population Health, Faculty of Health Sciences, American University of Beirut, Beirut, Lebanon, 8 Global Action for Public Health Services (GAPS), Buea, Cameroon, 9 Faculty of Medicine and Biomedical Sciences, University of Yaoundé I, Yaoundé, Cameroon, 10 Centre Pasteur of Cameroon, Yaoundé, Cameroon, 11 Krefting Research Centre, Institute of Medicine, University of Gothenburg, Gothenburg, Sweden, 12 Yale School of Public Health, New Haven, Connecticut, United States of America

* Libwea_j@yahoo.com, john.njumalibwea@tuni.fi

## Abstract

### Background

Respiratory syncytial virus (RSV) is one of the major pathogens frequently associated with severe respiratory tract infections in younger children and older adults globally. There is an unmet need with a lack of routine country-specific databases and/or RSV surveillance systems on RSV disease burden among adults in most low- and middle-income countries, including Cameroon. We aim to estimate the adult RSV burden needed to develop a framework for establishing an RSV surveillance database in Cameroon.

### Methods and analysis

A two-phase study approach will be implemented, including a literature review and a review of medical records. First, a systematic review of available literature will provide insights into the current burden of RSV in adults in Cameroon, searching the following databases: Global Health, PubMed, CINAHL, Embase, African Journal Online Library, Scopus, Global Index Medicus, Cochrane databases, and grey literature search. Identified studies will be included if they reported on the RSV burden of disease among Cameroonian adults aged ≥18 years from 1st January 1990 to 31st December 2023. A narrative synthesis of the evidence will be provided. A meta-analysis will be conducted using a random effect model, when feasible. Two co-authors will independently perform data screening, extraction, and synthesis and will be reported according to the PRISMA-P guidelines for writing systematic review

**Data Availability Statement:** No datasets were generated or analysed during the current study. All

relevant data from this study will be made available upon study completion.

**Funding:** The author(s) received no specific funding for this work.

**Competing interests:** The authors have declared that no competing interests exist.

protocols. Secondly, a retrospective cohort design will permit data analysis on RSV among adults in the laboratory registers at the National Influenza Center. Medical records will be reviewed to link patients' files from emanating hospitals to capture relevant demographic, laboratory, and clinical data. The International Classification of Diseases and Clinical Modifications 10th revision (ICD-10-CM) codes will be used to classify the different RSV outcomes retrospectively.

## Results

The primary outcome is quantifying the RSV burden among the adult population, which can help inform policy on establishing an RSV surveillance database in Cameroon. The secondary outcomes include (i) estimates of RSV prevalence among Cameroonian adult age groups, (ii) RSV determinants, and (iii) clinical outcomes, including proportions of RSV-associated morbidity and/or death among age-stratified Cameroonian adults with medically attended acute respiratory tract infections.

## Conclusions

The evidence generated from the two projects will be used for further engagement with relevant stakeholders, including policymakers, clinicians, and researchers, to develop a framework for systematically establishing an RSV surveillance database in Cameroon. This study proposal has been registered (CRD42023460616) with the University of York Center for Reviews and Dissemination of the International Prospective Register of Systematic Reviews (PROSPERO).

## 1. Introduction

Respiratory syncytial virus (RSV) remains one of the significant pathogens frequently associated with severe acute respiratory tract infections (ARI) in younger children and older adults globally [1–4]. A recent meta-analysis reported that RSV-associated hospitalization and ARI clinic visitation incidences were highest among young children [1]. Still, a substantial burden of ARI due to RSV was observed among older children and adults [1], especially elderly individuals and those with underlying medical conditions [4]. This may partly be explained by the fact that immunity to RSV is incomplete, and infections recur throughout life, leading to growing evidence that rates of healthcare utilization, hospitalization, morbidity, and mortality among adults with RSV infection may be similar to those observed with influenza infection [5, 6].

It is expected that adult RSV vaccines, which have recently been licensed in the USA with demonstrated moderate to high efficacy in preventing RSV-associated lower respiratory tract disease, will potentially prevent substantial morbidity and mortality among older adults [7]. However it may take a couple of years for these vaccines to reach resource-low settings where the recommended antimicrobial therapies for high-risk populations are costly and usually not widely available [8]. The development of adult RSV vaccines and their subsequent rollout will be a significant advancement in global public health, considering the high disease burden resulting from RSV [2, 9, 10]. However, to effectively quantify the expected long-term benefits of the vaccines, a comprehensive synthesis of the global adult RSV burden is needed to inform public health decision-making [1]. Although a global problem, while RSV disease surveillance systems and databases are known to be maintained in high-income economies, this remains

challenging for resource-low settings. Despite this, several studies that described the epidemiology of RSV infection in low- and middle-income countries (LMIC) have mainly focused on children [3, 9, 11]. The lack of a comprehensive RSV evidence base in adults has been caused by the absence of country-specific databases on RSV disease burden among adults and/or routine RSV surveillance systems in most LMICs, including Cameroon. This unmet need for more data that will subsequently contribute to setting the path for sustainable preventive strategies requires the utmost attention and strongly motivates this proposal.

Therefore, to address this gap, the overarching goal of the proposal is to quantify the adult RSV burden needed to support the framework that would help establish an RSV surveillance database in Cameroon. Hence, we aim to perform a comprehensive review of evidence on the epidemiology, determinants/risk factors, and clinical manifestations of RSV among Cameroonian adults ($\geq$ 18 years old) and to provide recommendations for the implementation of a national viral surveillance framework in the country. This is innovative considering that the methods for RSV surveillance may also be used for flu, SARS-CoV-2, as well as for human papillomavirus surveillance. When feasible, a critical appraisal of the evidence and meta-analysis will be performed, i.e., a meta-analysis will be performed if participants, interventions, comparisons, and outcomes are judged to be sufficiently similar to ensure a clinically meaningful answer [12]. Based on the evidence synthesis to be generated from the systematic review, a roadmap of what has been done on the topic in Cameroon will be ascertained, and then that knowledge will be integrated into the analysis of the retrospective hospital/laboratory register-based RSV adult data.

## 2. Study rationale

Although commonly associated with acute lower respiratory tract infections in young children, worldwide, RSV is also a major viral pathogen causing severe lung disease in the adult population, particularly in older adults [2]. With more data describing the RSV disease burden and associated costs globally, the need to introduce life-saving RSV vaccines will be optimal in reducing the pain, disability, and loss of human lives.

In general, surveillance systems aim to build an evidence base. Therefore, getting a better understanding of the RSV disease burden in older adults across the globe is a first good step. However, Cameroon's situation is even more challenging without a surveillance system. Moreover, the epidemiology of RSV adult infections in people with ARI has not yet been comprehensively investigated in Cameroon.

We strongly posit that the absence of an RSV surveillance system/database in Cameroon is an unmet need, and it is vital to establish one. However, before establishing one, we must seek to understand the local situation and use that knowledge to develop a surveillance system. Like most LMICs, Cameroon has a limited healthcare budget and presents a unique ecological context that could serve as a rich reservoir for RSV infection. Still, comprehensive knowledge about the epidemiology of RSV disease burden in adults in the country is lacking. Therefore, an extensive review is essential in appraising the current situation in Cameroon to generate evidence-based recommendations needed to support the implementation of an RSV surveillance system.

### 2.1. Review question

What are the prevalence, determinants (risk factors), and clinical outcomes of RSV infections among Cameroonian adults ($\geq$18 years old) who were hospitalized or consulted for ARI?

## 3. Aims and objectives

The study's overarching goal is to quantify the adult population's RSV burden which can help inform policy on formulating a framework to establish an RSV surveillance/database in Cameroon. The results from a systematic review and analysis of the retrospective hospital/laboratory RSV adult data will provide insights into the current burden of the disease and guide policymakers and other stakeholders toward establishing the RSV surveillance database in the country.

### 3.1. Specific objectives

These will include:

1. To estimate the prevalence of RSV among adults (≥18 years) in Cameroon who were hospitalized or consulted for ARI from 1st January 1990 to 31st December 2023;

2. To investigate the associated RSV determinants among Cameroonian adults;

3. To determine the clinical outcomes, including age-stratified proportions of RSV-associated morbidity and/or death among adults (≥18 years) in Cameroon who were hospitalized or consulted for ARI from 1st January 1990 to 31st December 2023;

4. To identify evidence-based recommendations that support sustainable engagement and resource sharing on best practices for establishing a national RSV surveillance/database programme implementation in Cameroon.

## 4. Materials and methods

A two-phase study design will be considered, i.e., a retrospective laboratory/hospital register-based cohort study design and a systematic review (Fig 1).

### 4.1. Ethical considerations and dissemination

Ethical approval will not be required since we will be using secondary data (and no patients or members of the public will be involved directly) for the systematic review arm. However, ethical clearance and administrative authorizations will be obtained from the required institutional review boards relating to the use of hospital/laboratory registered data for the retrospective cohort arm. We intend to submit the protocol and, subsequently, the ensuing findings of this review to peer-reviewed journals on RSV, as well as present our findings at national, regional, and international scientific meetings, conferences, and/or seminars as well as submit the results for publication in internationally peer-reviewed journals in the subject areas. This protocol has been registered (CRD42023460616) with the International Prospective Register of Systematic Reviews (PROSPERO).

### 4.2. Methodology for retrospective laboratory register-based cohort study

The absence of an RSV database or surveillance system in Cameroon necessitates that we conduct a retrospective laboratory/hospital register-based cohort study. In this phase of the proposed project, the inception cohort will be patients with RSV-associated laboratory data who were hospitalized or consulted for ARI contained in registers at the Virology laboratories (with PCR machines) of the National Influenza Center, Pasteur Institute, Yaoundé, Cameroon. Medical records will be reviewed to capture relevant demographic, laboratory, and clinical data. Pertinent information will be extracted and keyed into study-specific case report forms

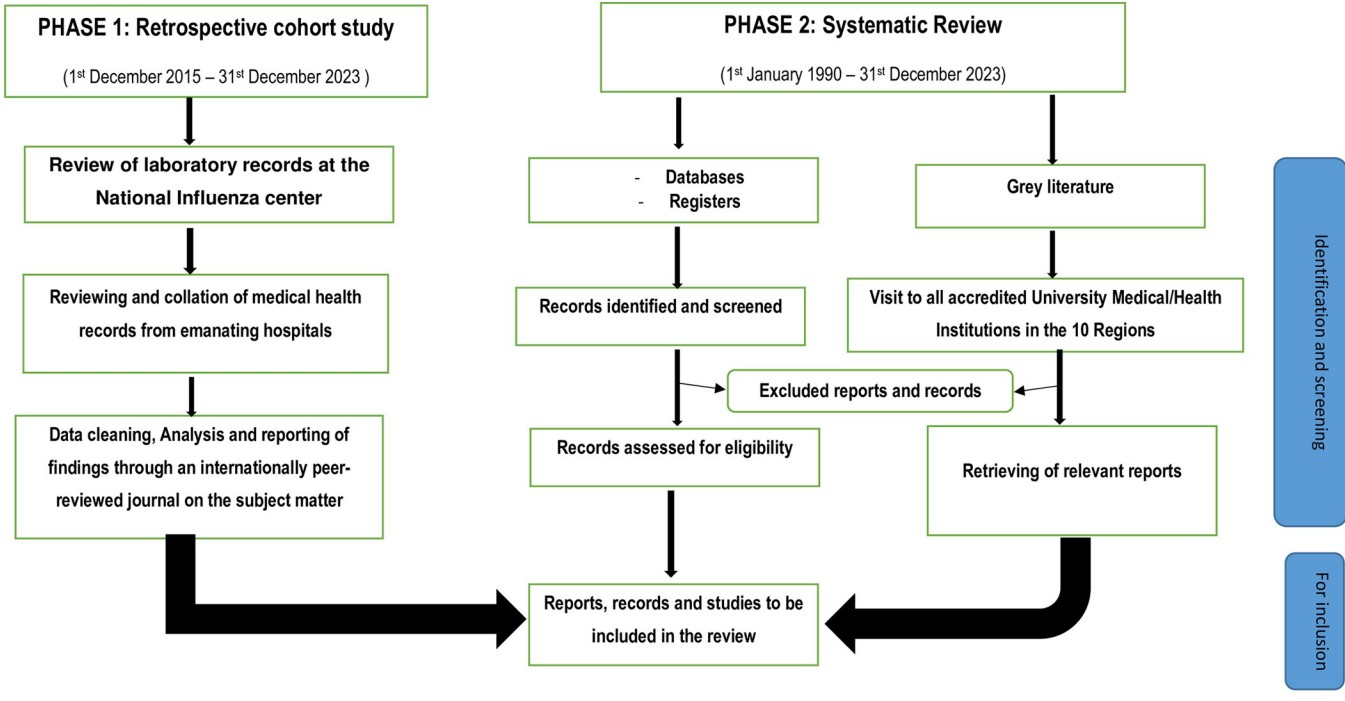

**Fig 1. Outline of study protocol and search strategy.**

(CRF). Eligible for inclusion were adults aged ≥18 years; residency in any community within Cameroon; availability of a referral form from an authorized physician or specialist working within any of the accredited hospital establishments in Cameroon; and registered cases must have occurred within the study period i.e., from 1st January 2015 to 31st December 2023 (The rationale for the selected period is to establish trends specific to RSV, a few years before and during the COVID-19 era, which may provide valuable insights for policy recommendations in preparedness and response measures to future pandemics). Those not meeting the eligibility mentioned above conditions will be excluded from the analyses. However, we will exceptionally enroll all RSV-ARI-associated cases as this will be an opportunity to get a wider understanding of the entire data. The main outcome measure will be prevalence (morbidity and/or mortality), which will be measured as a fraction of ARI among medically attended Cameroonian adults who tested positive for RSV, expressed in percentages.

Electronic health records are not generally maintained in Cameroon, so experienced study personnel will carefully use personal identifiers (including full names, telephone numbers, age, sex, family information, date of admission, name of consulting physician, and referral hospital) during the health records review process, as was previously done [13]. In addition, data on place of residence, cause of hospitalization (diagnosis at admission/outpatient visit), discharge diagnosis, length of hospital stay, underlying medical conditions, potential risk factors (e.g., history of chronic obstructive pulmonary disease, COVID-19, stroke, diabetes, immunosuppression, central nervous system, kidney, or liver disorders) as well as on socio-economic status will be captured. Data from the pre-COVID-19 era (2015–2019) will be compared with that of the post-COVID-19 era (2020–2023). The International Classification of Diseases-Tenth Revision (ICD-10) codes for RSV (B97.4, J12.1, J20.5, J21.0) and bronchiolitis (RSV codes plus J21.8, J21.9) will be used to classify the different RSV-associated morbidity and/or death among adults (≥18 years) in Cameroon who were hospitalized or consulted for ARI.

Since RSV is reported as one of several causes of bronchiolitis [14], it will be defined by one of the RSV codes or one of the following codes: acute bronchiolitis due to other specified organisms (J21.8); or acute bronchiolitis, unspecified (J21.9).

## 4.3. Methodology for systematic review

In addition to the retrospective laboratory register-based cohort study, is a systematic review arm of the protocol. The review will be conducted and reported following the Preferred Reporting Items for Systematic Reviews (PRISMA) guidelines [15, 16].

**4.3.1. Criteria for inclusion and exclusion.** We will search for randomized control trials and observational studies (including cohort, cross-sectional, and case-control), systematic reviews (for reviewing and extraction of individual studies), and conference abstracts that describe the prevalence, risk factors, and clinical outcomes of medically attended RSV among Cameroonian adults (≥18 years). Risk factors will include a history of chronic obstructive pulmonary disease, stroke, diabetes, immunosuppression, central nervous system, kidney, or liver disorders, as previously reported [1]. Clinical outcomes will include a history of underlying conditions, e.g., pulmonary and cardiovascular diseases, presenting signs or symptoms (e.g., cough, sore throat, breathing difficulties, hypoxemia, fever≥38˚C), discharge diagnosis of lower respiratory tract infections (e.g., pneumonia, bronchiolitis, and bronchitis), chest radiography finding (e.g., hyper aeration, pneumonia, including ≥7days hospitalization), intensive care unit (ICU) admission, mechanical ventilation, discharged alive, transferred/referral or death [17–19]. Also, we will include studies that considered patients with a clinical diagnosis of acute respiratory tract infection as defined in each study and demonstrate the diagnostic algorithm and techniques used for the RSV sampling method, clinical signs/outcomes, and advocacy for RSV surveillance system/database establishment in Cameroon.

To be excluded are studies conducted among adult populations reported during an outbreak period, case reports, or studies lacking extractable primary data and/or explicit method descriptions. However, if some data are missing from eligible primary studies, the authors of such literature will be contacted. Additionally, there will be no language restrictions as we will search for all relevant literature or studies published from 1st January 1990 to 31st December 2023 that were conducted in or reported on data from Cameroon on RSV disease burden in adults.

**4.3.2. Source of information.** A comprehensive search of available literature will be performed using the following electronic databases: Global Health, PubMed, CINAHL, Embase, African Journal Online Library, Scopus, Global Index Medicus, and Cochrane databases, in addition to the grey literature search. Identified studies will be included if they reported on the RSV burden of disease among Cameroonian adults aged 18 years or above from 1st January 1990 to 31st December 2023. Moreover, the search for grey literature will be performed through sources such as ProQuest Dissertations and Theses Global, Mascot/Wotro, Effective Public Health Practice Projects, Public Health Grey Literature Sources, and Health Evidence as well as physically visiting the libraries of the ten (10) medical faculties and health institutions nation-wide to identify, access and retrieve any unpublished works on RSV.

**4.3.3. Search strategies.** An electronic systematic article search will be performed using the aforementioned electronic databases for relevant studies based on the research question: "What are the determinants/risk factors, clinical outcomes, and prevalence of RSV infections among Cameroonian adults (≥18 years) needed to engage policymakers for the establishment of an RSV surveillance system in Cameroon"?

The following concepts will be used:

**Concept 1**: Respiratory syncytial virus (RSV) surveillance system (database)

☐ ("respiratory syncytial viruses"[MeSH Terms] OR "Respiratory Syncytial Virus") AND RSV AND (("epidemiology" OR "epidemiology"[MeSH Terms] OR surveillance) AND system)) OR database

**Concept 2**: <u>Determinants or Risk factors of Respiratory syncytial virus (RSV) infections</u>

☐ Determinants OR (("risk factors" [MeSH Terms] OR "Risk factors") AND "respiratory syncytial virus infections" [MeSH Terms])

**Concept 3**: <u>Clinical outcomes (manifestations) Respiratory syncytial virus (RSV) infections</u>

☐ Clinical AND Outcomes OR manifestations

**Concept 4**: <u>Prevalence Respiratory syncytial virus (RSV) infections</u>

☐ Prevalen* OR "epidemiology" OR "prevalence" [MeSH Terms] OR prevalence

**Concept 5**: <u>Adults (≥18 years old)</u>

☐ "adult"[MeSH Terms] OR Adults

**Concept 6**: <u>Cameroon</u>

☐ "Cameroon"[MeSH Terms] OR Cameroon

**4.3.4. Screening of retrieved literature.** The literature retrieved from the databases will be transferred to Endnote to remove duplicate papers. After that, the papers will be exported to Rayyan for further screening, and each title and/or abstract will be screened independently by at least two co-authors for potentially eligible studies. The full texts of selected papers will be obtained for screening, and if discrepancies arise between the two co-authors, they will be resolved through discussion. If no consensus is reached, a third co-author will arbitrate. The PRISMA flow chart will be used to report the screening process.

**4.3.5. Registration and reporting.** This study proposal has been registered (CRD42023460616) with the University of York Center for Reviews and Dissemination of the International Prospective Register of Systematic Reviews (PROSPERO). The protocol is reported according to the PRISMA-P guidelines for writing systematic review protocols [16].

**4.3.6. Data extraction.** A study-specific data extraction form will be developed using Microsoft Excel. At least two co-authors or reviewers will independently use it to extract relevant study data from eligible studies. Before full use with all included studies, the data extraction form will first be piloted using a couple of the selected studies. Following the piloting, necessary amendments will be made to the extraction form to ensure that it is suitable for capturing all relevant data from eligible studies. Information collected from each eligible study will include the title, year of publication, authors, study design, description of the RSV surveillance/database framework agenda, and reports on the prevalence, drivers and/or clinical outcomes as outlined in the study objectives.

**4.3.7. Quality assessment.** Quality assessment of included studies will be performed independently by at least two co-authors using the Cochrane Risk of Bias tool for randomized controlled trials and the Newcastle-Ottawa Quality Assessment Scale for nonrandomized studies [20, 21]. Disagreement between the co-authors will be resolved by discussion, with the involvement of a third co-author for a final opinion, if needed.

**4.3.8. Data synthesis.** A descriptive summary of the characteristics of the included studies and the main findings will be provided. For the risk factors, we will independently compare the measures of association (Odds Ratio or Risk Ratio) for each study design. Depending on

the characteristics of the included studies, results will be stratified according to various adult age groups to visualize the RSV disease burden among older adults (≥65 years) and other adults.

Further, heterogeneity between studies will be evaluated by considering variability between the individual study characteristics, including participants' selection, risk factor assessment, clinical outcome assessment, loss to follow-up in cohort studies, non-response, and missing data. The Chi-square ($X^2$) test and I-squared statistics will be used to estimate statistical heterogeneity at the 5% significance level. Heterogeneity will be classified based on the threshold suggested in the Cochrane Handbook for systematic reviews (0% to 40% = low; 30% to 60% = moderate heterogeneity; 50% to 90% = substantial heterogeneity, and 75% to 100% = considerable heterogeneity). In an event where a high heterogeneity between the studies is observed, a detailed analysis of the design and characteristics of the individual studies will be performed to identify the sources of heterogeneity, and sensitivity analysis will be performed. When necessary, a meta-analysis will be performed using a random effect model, and estimates will be reported with their 95% confidence intervals. The meta-analysis will be carried out using RevMan5.4 according to the guidelines noted in the Cochrane Handbook, and results will be summarized in a forest/funnel plot.

**4.3.9. Expected study outcomes and prioritization.**   Based on the study objectives, the primary outcome will be to quantify the RSV burden among the adult population, which is needed to support evidence-based recommendations for the establishment of a national RSV database/surveillance system in Cameroon. The secondary outcomes will include (i) estimates of the prevalence of RSV among adult age groups in Cameroon, (ii) RSV determinants or risk factors, and (iii) clinical outcomes, including proportions of RSV-associated morbidity and/or death among Cameroonian adults with medically attended acute respiratory tract infections, stratified by adult age groups.

## 5. Discussion

This, to the best of our knowledge, is the first systematic review study protocol on the epidemiology, risk factors /determinants, and clinical manifestations of RSV among Cameroonian adults (≥ 18 years) needed to support recommendations on establishing a national RSV surveillance network and best practices for policy implementation in Cameroon. The different types of studies targeting the Cameroonian adult population with medically attended acute respiratory tract infection, risk factors, and clinical outcomes have been comprehensively described in accordance with the research question. The data sources, search strategy, data extraction, methodological quality of the studies, data synthesis approach, risk of bias assessment, and reporting have been described following the PRISMA guidelines [16]. Moreover, using the "Population/Intervention (Exposure)/Comparison (Comparator)/Outcome (PICO/PEO)" concept [22, 23], the research question for this review has been conceptualized to ensure a robust and comprehensive search of the available relevant literature.

The findings are expected to provide evidence on best practices and resource sharing for policy consideration in establishing an RSV surveillance (database) for decision-makers, government authorities, healthcare providers, and other stakeholders at the local and international levels. Additionally, the study seeks to establish determinants, prevalence, and clinical outcomes of RSV disease burden specific to older adults (≥65 years) and across the adult age group. More so, because the National Influenza Laboratory does not maintain a separate database for the young or adult population, it gives a singular opportunity to access all available data. This provides a better understanding of the disease epidemiology across the entire population (including adults ≥18 years). Hence, the findings will present a vivid picture of what

strata of the adult population are most at risk and provide valuable insights for health policy prioritization of possible preventive strategies and/or intervention, including antimicrobials and vaccine rollout to targeted age groups, whenever adult RSV vaccination programme in the country begins.

Despite our intention to conduct a comprehensive review, we are conscious of possible challenges, such as age differences in clinical presentations between RSV-positive and RSV-negative adults with acute respiratory tract infection, diagnostic methods (including testing practices), healthcare-seeking behaviour, case definitions, and coding systems, which may contribute to potential sources of bias in our findings. However, in using the PRISMA guidelines [16], we remain confident this will demonstrate a high degree of transparency of the study processes from initiation to study design, screening of eligible studies, data extraction and synthesis unto the reporting and dissemination of findings, which will engage policymakers and government towards establishing the needed RSV surveillance system.

## 6. Conclusion

The RSV disease burden among the adult population, especially the elderly, remains a global public health threat and needs joint efforts to sustainably address the scourge at both local, national, and international levels. Regrettably, the absence of an RSV surveillance platform in Cameroon is an unmet need for more data and sustainable preventive strategies that must be addressed for the country to align with its developmental agenda. The expected findings from this study will include quantification or estimates of RSV prevalence among Cameroonian adult age groups, determinants of RSV, and clinical outcomes (e.g., proportions of RSV-associated morbidity and/or death among age-stratified Cameroonian adults with medically attended ARI). RSV is one of the significant pathogens frequently associated with severe respiratory tract infections in younger children and elderly adults globally, but adult disease burden data from Cameroon is lacking. Moreover, healthcare costs in Cameroon are traditionally borne through out-of-pocket payments, but epidemic-prone diseases that are under national surveillance benefit from the government's subventions. We are confident that the findings from this study will provide the evidence base that will drive advocacy and inform policy on establishing an RSV surveillance database in the country. Therefore, understanding the epidemiology, risk factors, and clinical outcomes of RSV disease burden in Cameroon will harness evidence-based findings needed to establish a national RSV surveillance system before the advent of RSV adult vaccinations and make further additions to the publicly available knowledge about RSV disease epidemiology.

## Supporting information

**S1 Checklist. PRISMA-P 2015 checklist.**
(DOCX)

## Acknowledgments

We do express our gratitude to all authors whose works have been referenced in the write-up of this paper. Special appreciations are extended to all collaborating authors for their time and effort in completing this work with no financial support.

## Author Contributions

**Conceptualization:** John Njuma Libwea, Linda Esso, Andreas Ateke Njoh, Che Henry Ngwa, Bright I. Nwaru.

**Data curation:** John Njuma Libwea, Bright I. Nwaru.

**Investigation:** John Njuma Libwea, Richard Njouom.

**Methodology:** John Njuma Libwea, Andreas Ateke Njoh, Che Henry Ngwa, Chanceline Bilounga Ndongo, Bright I. Nwaru, Dan Weinberger.

**Project administration:** John Njuma Libwea, Linda Esso, Vivienne Ngomba Armelle, Chanceline Bilounga Ndongo, Georges Alain Etoundi Mballa, Richard Njouom, Sinata Koulla-Shiro.

**Resources:** John Njuma Libwea, Che Henry Ngwa, Vivienne Ngomba Armelle, Chanceline Bilounga Ndongo, Georges Alain Etoundi Mballa, Richard Njouom, Sinata Koulla-Shiro.

**Software:** John Njuma Libwea, Vivienne Ngomba Armelle, Georges Alain Etoundi Mballa.

**Supervision:** John Njuma Libwea, Linda Esso, Vivienne Ngomba Armelle, Chanceline Bilounga Ndongo, Georges Alain Etoundi Mballa, Bright I. Nwaru, Dan Weinberger, Richard Njouom, Sinata Koulla-Shiro.

**Validation:** John Njuma Libwea, Linda Esso, Andreas Ateke Njoh, Che Henry Ngwa, Vivienne Ngomba Armelle, Chanceline Bilounga Ndongo, Georges Alain Etoundi Mballa, Bright I. Nwaru, Dan Weinberger, Richard Njouom, Sinata Koulla-Shiro.

**Visualization:** John Njuma Libwea, Linda Esso, Andreas Ateke Njoh, Che Henry Ngwa, Vivienne Ngomba Armelle, Chanceline Bilounga Ndongo, Georges Alain Etoundi Mballa, Bright I. Nwaru, Dan Weinberger, Richard Njouom, Sinata Koulla-Shiro.

**Writing – original draft:** John Njuma Libwea, Andreas Ateke Njoh, Georges Alain Etoundi Mballa, Bright I. Nwaru.

**Writing – review & editing:** John Njuma Libwea, Linda Esso, Andreas Ateke Njoh, Che Henry Ngwa, Vivienne Ngomba Armelle, Chanceline Bilounga Ndongo, Bright I. Nwaru, Dan Weinberger, Richard Njouom, Sinata Koulla-Shiro.

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
