## [Decision Letter · Decision Letter 0]

1 Mar 2024

PONE-D-23-35302Generating the evidence to support the establishment of a Respiratory Syncytial Virus surveillance system in Cameroon: A study  protocolPLOS ONE

Dear Dr. Njuma Libwea,

Thank you for submitting your manuscript to PLOS ONE. After careful consideration, we feel that it has merit but does not fully meet PLOS ONE’s publication criteria as it currently stands. Therefore, we invite you to submit a revised version of the manuscript that addresses the points raised during the review process. Please submit your revised manuscript by Apr 15 2024 11:59PM. If you will need more time than this to complete your revisions, please reply to this message or contact the journal office at plosone@plos.org. Please include the following items when submitting your revised manuscript:A rebuttal letter that responds to each point raised by the academic editor and reviewer(s). You should upload this letter as a separate file labeled 'Response to Reviewers'.A marked-up copy of your manuscript that highlights changes made to the original version. You should upload this as a separate file labeled 'Revised Manuscript with Track Changes'.An unmarked version of your revised paper without tracked changes. You should upload this as a separate file labeled 'Manuscript'.

We look forward to receiving your revised manuscript.

Kind regards,

Dong Keon Yon, MD, FACAAI, FAAAAI

Academic Editor

PLOS ONE

Journal Requirements:

Additional Editor Comments:

Thank you for submitting your manuscript. The reviewers and I believe it is of potential value for our readers. However, the reviewers have raised a number of very important issues, and their excellent comments will need to be adequately addressed in a revision before the acceptability of your manuscript for publication in the Journal can be determined. We cannot guarantee that your revised paper will be chosen for publication; this would be solely based on how satisfactorily you have addressed the reviewer comments.

# The protocol is reported according to the PRISMA-P guidelines for writing systematic review protocols [10]. -> Please additionally cite the PRISMA 2020 guideline (DOI: https://doi.org/10.54724/lc.2022.e9).

Reviewers' comments:

Reviewer's Responses to Questions

**Comments to the Author**

1. Does the manuscript provide a valid rationale for the proposed study, with clearly identified and justified research questions?

Reviewer #1: Partly

Reviewer #2: Yes

2. Is the protocol technically sound and planned in a manner that will lead to a meaningful outcome and allow testing the stated hypotheses?

Reviewer #1: Partly

Reviewer #2: Yes

3. Is the methodology feasible and described in sufficient detail to allow the work to be replicable?

Reviewer #1: No

Reviewer #2: Yes

4. Have the authors described where all data underlying the findings will be made available when the study is complete?

Reviewer #1: No

Reviewer #2: Yes

5. Is the manuscript presented in an intelligible fashion and written in standard English?

Reviewer #1: Yes

Reviewer #2: No

6. Review Comments to the Author

You may also provide optional suggestions and comments to authors that they might find helpful in planning their study.

Reviewer #1: PLOS Review 2023Dec10.RSV surveillance Cameroon

This paper describes plans for studies to determine the burden of Respiratory Syncytial Virus (RSV) in adults in Cameroon. The overarching goal is to generate evidence to support a RSV surveillance system for RSV in that country.

In the first phase a literature review will be done to determine the burden of illness of RSV in Cameroon. In the second phase a retrospective search for RSV isolates (reviewer paraphrase) identified in the laboratory register of the National Influenza Center will be obtained, and a hand search of patient Health Records located at hospitals associated with these virus isolations will be done. Data will be collected to describe the clinical course and patient demographics.

General comments:

RSV is recognized as an important pathogen for children, older adults and persons with certain co-morbidities. As there are preventive and therapeutic interventions increasingly available, it is important to understand country-level disease burden to inform health decision making about feasibility and cost of these interventions in one’s own setting.

The paper is describing what the investigators intend to do. This reviewer finds that there is insufficient detail in the current version of the paper to assess if the protocol is feasible or methodologically appropriate.

Specific comments:

Abstract

1. Goal of the study – Is it the purpose establish a surveillance system or determine if there is enough available data to evaluate if a surveillance system is needed to know the burden of disease?

2. What is the status of surveillance for infectious diseases in Cameroon, and of public health? Are there reportable diseases and feedback to health care providers and knowledge users about other communicable diseases? Is this endeavour starting from scratch or building on a strong foundation in which another virus will be added to an existing system?

3. Presumably the authors have done a quick landscape analysis to see if there are any peer-reviewed publication on RSV from Cameroon.

4. Methods: Medical records will be “explored”. Much more precise language is needed.

5. How is RSV burden defined. Are the authors looking at the severe end of the spectrum only (patients sick enough to be hospitalized and with access to care- i.e. would personal resources limit the likelihood of testing or of medical care?)

Introduction

1. Line 57. There are authorized vaccines/passive immunization products, add references as wording implies these are in late stage development.

2. There are no recommended therapies. (i.e. treatments)

3. Line 76. …when feasible a critical appraisal of the evidence…will be performed”. How could this step be justifiably not done?

4. Line 84 “an ever increasing disease burden” Is the burden increasing or are there just more data describing the burden

5. Line 97. What does "consulted for RSV" mean?

6. Suggest using epidemiologic terms that are generally understood, or define the terms when first used. For example the term “driver” is used, line 10. Is this a risk factor or a predictor. “Driver” seems to imply causal association.

Methods (4)

1. Is the intent to determine prevalence? What is the denominator?

2. Line 131/ personnel will carefully use personal identifiers (…full names, phone numbers…etc) There is is a threat to confidentiality and privacy. More details of measures to mitigate risks are needed. Have the investigators had preliminary conversations with authorities to assess feasibility of this design.

3. The study is described as a cohort study. Line 148 Please provide evidence that this is the case (what is the inception cohort, who is eligible, ineligible, how will outcome measures be ascertained etc)

The writing is sometimes wordy and repetitive and could be condensed.

Reviewer #2: -This research is short but shows great effort in demonstrating the medical situation of RSV in Cameroon. It offers evidence and great insight that may be used in future policy-making and further research in relevant fields. However, some concerns must be demonstrated by authors before publication.

-There are several style errors detected at first glance. English proofreading is vital.

-Information about RSV and ARI ought to be specified in the introduction.

-The authors assess the diffusion of innovation of this research in that the methods may be used for other contagious diseases such as SARS-CoV-2, flu, and human papillomavirus surveillance. Is there a specific reason the authors focused on RSV instead of other diseases?

-I would like to recommend the authors make a figure of the procedure of the search strategy to enhance the readability of readers.

-The authors wanted to discuss the burden of the adult population due to RSV. Considering that this study excluded young children, the main population suffering from RSV infection, is there a way to exclude the other major population, aged 65 and older, who suffer from RSV infection?

-The conclusion seems somewhat vague. Please specify the expected results in the conclusion.

-The paper needs more citations for the credibility of the study protocol.

- The authors discuss ways to address possible heterogeneity between studies. If heterogeneity occurs, what is the main cause of heterogeneity?

- Please define the risk factors in a more specified manner.

7. PLOS authors have the option to publish the peer review history of their article (what does this mean?). If published, this will include your full peer review and any attached files.

Reviewer #1: No

Reviewer #2: No

---

## [Author Response · Author response to Decision Letter 0]

12 Apr 2024

Reviewer #1: PLOS Review 2023Dec10.RSV surveillance Cameroon

This paper describes plans for studies to determine the burden of Respiratory Syncytial Virus (RSV) in adults in Cameroon. The overarching goal is to generate evidence to support a RSV surveillance system for RSV in that country.

In the first phase a literature review will be done to determine the burden of illness of RSV in Cameroon. In the second phase a retrospective search for RSV isolates (reviewer paraphrase) identified in the laboratory register of the National Influenza Center will be obtained, and a hand search of patient Health Records located at hospitals associated with these virus isolations will be done. Data will be collected to describe the clinical course and patient demographics.

General comments:

RSV is recognized as an important pathogen for children, older adults and persons with certain co-morbidities. As there are preventive and therapeutic interventions increasingly available, it is important to understand country-level disease burden to inform health decision making about feasibility and cost of these interventions in one’s own setting.

The paper is describing what the investigators intend to do. This reviewer finds that there is insufficient detail in the current version of the paper to assess if the protocol is feasible or methodologically appropriate.

Response to General comments:

Specific comments:

Abstract

Comment no.1: Goal of the study – Is it the purpose establish a surveillance system or determine if there is enough available data to evaluate if a surveillance system is needed to know the burden of disease?

Response no. 1: We thank the Reviewer for this question. The goal of the study is to quantify the adult RSV burden needed to support the framework that would help establish an RSV surveillance database in Cameroon. RSV burden is rapidly increasing across all age groups, especially within the infant and elderly adult populations as observed from laboratory diagnoses of samples obtained from patients with influenza like illnesses (ILI), sent to the National Influenza laboratory. Unfortunately, RSV is somehow being neglected as it is not among the Epidemic-prone diseases under routine surveillance in the country. And as evidence demonstrates high levels with the infant population (Staadegaard L et al., 2021), the RSV burden of disease in the adult population may even be higher. And with the envisaged implementation of adult RSV vaccine in the future, scientific documentation of a baseline disease burden is paramount.

Comment no. 2: What is the status of surveillance for infectious diseases in Cameroon, and of public health? Are there reportable diseases and feedback to health care providers and knowledge users about other communicable diseases? Is this endeavour starting from scratch or building on a strong foundation in which another virus will be added to an existing system?

Response no. 2: This is a very vital question. Here, we will say that the endeavour is not starting from scratch as since year 2007, “Centre Pasteur Cameroon (CPC)” was designated National Influenza Centre (NIC) by the Ministry of Public Health, Cameroon. As from then, surveillance of influenza and most recently, COVID-19 is being performed routinely, except RSV as earlier stated in Comment no.1 above. However, numerous published studies have led to better understanding of epidemiology of major respiratory pathogens in the country, including RSV (Kengne-Nde C et al., 2020; Kenmoe S et al., 2024). Hence, the need to comprehensively quantify these literatures needed to establish a national routine surveillance for RSV, especially in the advent of RSV adult vaccine programme implementation.

Comment no. 3: Presumably the authors have done a quick landscape analysis to see if there are any peer-reviewed publication on RSV from Cameroon.

Response no. 3: Yes, this has been done and the peer-reviewed publications have been conducted by some of the co-authors of the present manuscript including Prof. Richard Njoum who heads the NIC laboratory.

Comment no. 4: Methods: Medical records will be “explored”. Much more precise language is needed.

Response no. 4: We thank the Reviewer for drawing our attention to this. It has been modified and the sentence now reads, “… medical records will be reviewed…”

Comment no. 5: How is RSV burden defined. Are the authors looking at the severe end of the spectrum only (patients sick enough to be hospitalized and with access to care- i.e. would personal resources limit the likelihood of testing or of medical care?)

Response no. 5: We appreciate the Reviewer for this question as it aligns with one of the justifications for this study and why RSV surveillance is an unmet need. Truly, the likelihood of testing and/or medical care are limited because cost of healthcare is borne through out-of-pocket payments, and this is a heavy toll to both the individual and the family. As such we will be looking for both hospitalizations and out-patients’ visits. Further, in this study, the International Classification of Diseases-Tenth Revision (ICD-10) codes for RSV (B97.4, J12.1, J20.5, J21.0) and bronchiolitis (RSV codes plus J21.8, J21.9) will be used to classify/define the different RSV-associated morbidity and/or death among adults (≥18 years) in Cameroon who were hospitalized or consulted for acute respiratory tract infections (ARI) and/or influenza-like illnesses (ILI). Since RSV is reported as one of several causes of bronchiolitis (Riccio MD et al., 2023), it will be defined by one of the RSV codes or one of the following codes: acute bronchiolitis due to other specified organisms (J21.8); or acute bronchiolitis, unspecified (J21.9).

Introduction

Comment no.1: Line 57. There are authorized vaccines/passive immunization products, add references as wording implies these are in late stage development.

Response no. 1: The line has been modified with the following text “It is expected that adult RSV vaccines, which have recently been licensed in the USA with demonstrated moderate to high efficacy in preventing RSV-associated lower respiratory tract disease, will potentially prevent substantial morbidity and mortality among older adults [7]. However, it may take a couple of years for these vaccines to reach resource-low settings where the recommended antimicrobial therapies for high-risk populations are costly and usually not widely available [8].” (see lines 58 - 64)

Comment no. 2: There are no recommended therapies. (i.e. treatments)

Response no. 2: We do respect the opinion of the Reviewer but we think there is literature on recommended treatment options for RSV e.g., Gatt D., et al. 2023. Prevention and Treatment Strategies for Respiratory Syncytial Virus (RSV). Pathogens. 2023 Jan 17;12(2):154. doi: 10.3390/pathogens12020154. PMID: 36839426; PMCID: PMC9961958.

Comment no. 3: Line 76. …when feasible a critical appraisal of the evidence…will be performed”. How could this step be justifiably not done?

Response no. 3: We appreciate the Reviewer’s question here and we will clarify that this statement draws inference from Cochrane’s Training manual on the pre-requisite for conducting meta-analysis (https://training.cochrane.org/handbook/current/chapter-10). “…Meta-analysis should only be considered when a group of studies is sufficiently homogeneous in terms of participants, interventions and outcomes to provide a meaningful summary…” This explains why we had stated that “When feasible, a critical appraisal of the evidence and meta-analysis will be performed”. Otherwise, if the studies found are not sufficiently homogenous then, ONLY the systematic review findings will be reported. Therefore, “when feasible, a critical appraisal of the evidence and meta-analysis will be performed, i.e., a meta-analysis will be performed if participants, interventions, comparisons, and outcomes are judged to be sufficiently similar to ensure a clinically meaningful answer [12]” (see lines 81-83).

Comment no. 4: Line 84 “an ever increasing disease burden” Is the burden increasing or are there just more data describing the burden

Response no. 4: We thank the Reviewer for drawing our attention to this. The sentence has been rephrased in the text and now reads “…With more data describing RSV disease burden and associated costs globally, the need to introduce life-saving RSV vaccines will be optimal in reducing the pain, disability, and loss of human lives” (See lines 90 -91).

Comment no. 5: Line 97. What does "consulted for RSV" mean?

Response no. 5: It reads “To estimate the prevalence of RSV among adults (≥18 years) in Cameroon who were hospitalized or consulted for ARI from January 1st 1990 to December 31st 2023” (see lines 114 - 115). i.e., RSV stands for Respiratory Syncytial Virus and ARI stands for Acute Respiratory tract Infections.

Comment no. 6: Suggest using epidemiologic terms that are generally understood, or define the terms when first used. For example, the term “driver” is used, line 10. Is this a risk factor or a predictor? “Driver” seems to imply causal association.

Response no. 6: Dear Reviewer, the term “driver” refers to determinant or risk factors. And as you have suggested to use epidemiologic terms that are common, drivers will be replaced with determinants/risk factors throughout the text.

Methods (4)

Comment no.1: Is the intent to determine prevalence? What is the denominator?

Response no. 1: Yes, the intent is to determine prevalence, which will be measured as a proportion of the denominator defined as all acute respiratory infections (ARI) and influenza-like illnesses (ILI) among medically attended Cameroonian adults who tested positive for RSV within the study period, expressed in percentages.

Comment no. 2: Line 131/ personnel will carefully use personal identifiers (…full names, phone numbers…etc.) There is a threat to confidentiality and privacy. More details of measures to mitigate risks are needed. Have the investigators had preliminary conversations with authorities to assess feasibility of this design.

Response no. 2: We thank the Reviewer on the emphasis on confidentiality. Yes, we will again use this opportunity to assure the Reviewer that all necessary measures have been taken to ensure that confidentiality and privacy are maintained throughout the process. For example, all the study personnel to be deployed are nurses within Centre Pasteur and/or collaborating hospital establishments. Secondly, we have worked in related studies previously and most importantly, a training seminar will be organized with all study personnel on Good Clinical Practices, to ensure that everyone is at the same level. Otherwise, most of the co-authors are designated authorities within the Ministry of Health deployed to the department of disease control, the national immunization programme, the national public health laboratory as well as Pasteur Institute. Definitely, there should be no major concern as the Team is a very solid one.

Comment no. 3: The study is described as a cohort study. Line 148 Please provide evidence that this is the case (what is the inception cohort, who is eligible, ineligible, how will outcome measures be ascertained etc). The writing is sometimes wordy and repetitive and could be condensed.

Response no. 3: Dear Reviewer, we will like to clarify here that, the study is in two phases with the first described as a retrospective cohort and the second, a systematic review. Information on eligibility and outcome measures were earlier provided for each study arm, respectively.

- For the first phase (see lines 136 - 151), the inception cohort includes patients with RSV-associated laboratory data who were hospitalized or consulted for ARI contained in registers at the Virology laboratories (with PCR machines) of the National Influenza Center, Pasteur Institute, Yaoundé, Cameroon.

- Inclusion criteria:

• Age: Eligible for inclusion are adults aged ≥18 years;

• Residency: Residing in any community within Cameroon;

• Having a referral from an authorized physician or specialist working within any of the accredited hospital establishments in Cameroon;

• Period: Registered cases must have occurred within the study period i.e., from January 1st 2015 to December 31st 2023

- Exclusion criteria:

• Those not meeting the aforementioned eligibility conditions will be excluded from the analyses. However, we will exceptionally enrol all RSV-ARI associated cases as this will be an opportunity to get a wider understanding of what this huge data contains, and later perform stratification to get a vivid picture across the life course i.e., different age groups.

- Outcome ascertainment: 

• The main outcome measure will be prevalence, which will be measured (evaluated) as a fraction (proportion) of acute respiratory infections (ARI) among medically attended Cameroonian adults who tested positive for RSV, expressed in percentages.

Reviewer #2: -This research is short but shows great effort in demonstrating the medical situation of RSV in Cameroon. It offers evidence and great insight that may be used in future policy-making and further research in relevant fields. However, some concerns must be demonstrated by authors before publication.

Comment no. 1: There are several style errors detected at first glance. English proofreading is vital.

Response no. 1: We thank the Reviewer for this commendation which has been considered.

Comment no. 2: Information about RSV and ARI ought to be specified in the introduction.

Response no. 2: Except otherwise, we think this information was earlier mentioned (see lines 52 – 59).

Comment no. 3: The authors assess the diffusion of innovation of this research in that the methods may be used for other contagious diseases such as SARS-CoV-2, flu, and human papillomavirus surveillance. Is there a specific reason the authors focused on RSV instead of other diseases?

Response no. 3: We thank the Reviewer for bringing this up as well. It is important to state here that the motivation for this has stemmed from field experience as the entire team have benefited from some advanced training in epidemiology and are either present or previous members of the Program for Monitoring Emerging Diseases (ProMED). Therefore, they have lots of experience in identifying unusual health events related to emerging and re-emerging infectious diseases and toxins affecting humans, animals and plants. RSV currently ticks that box, especially with the marketing of the recently approve RSV vaccines, we felt it was important to avoid earlier mistakes wherein, vaccination programmes have been implemented without a robust disease surveillance system. Also, with an undocumented country-specific adult RSV disease burden, it is suggested the adult population remains a high reservoir and potential source of community transmission of the pathogen (Cheryl Cohen et al., 2023). These issues surrounding the need for RSV surveillance system have been discussed during IDRS weekly meetings, yet policy lacked comprehensive evidence to justify government’s spending due to very tight budgetary constraints. Hence, the rationale of the study is to generate this evidence and use it to engage policy makers and other stakeholders on the need of establishing a routine surveillance for RSV nation-wide. 

Comment no. 4: I would like to recommend the authors make a figure of the procedure of the search strategy t Information about RSV and ARI o enhance the readability of readers.

Response no. 4: Dear Reviewer, thanks for the recommendation and please, find attached our suggested figure. However, an automated and more detailed version will be generated during the reporting of the study findings.

Comment no. 5: The authors wanted to discuss the burden of the adult population due to RSV. Considering that this study excluded young children, the main population suffering from RSV infection, is there a way to exclude the other major population, aged 65 and older, who suffer from RSV infection?

Response no.5: This is another very important concern the Reviewer has raised here. We will clarify here that the focus on this study is essentially on the adult population aged ≥ 18 years. However, it’s a syst

---

## [Decision Letter · Decision Letter 1]

25 Apr 2024

Generating the evidence to support the establishment of a Respiratory Syncytial Virus surveillance system in Cameroon: A study  protocol

PONE-D-23-35302R1

Dear Dr. Njuma Libwea,

We’re pleased to inform you that your manuscript has been judged scientifically suitable for publication and will be formally accepted for publication once it meets all outstanding technical requirements.

Kind regards,

Dong Keon Yon, MD, FACAAI, FAAAAI

Academic Editor

PLOS ONE

Additional Editor Comments (optional):

This is an excellent paper. During production stage, please improve the quality and DPI of Figure 1.

Reviewers' comments:

Reviewer's Responses to Questions

**Comments to the Author**

1. Does the manuscript provide a valid rationale for the proposed study, with clearly identified and justified research questions?

Reviewer #2: Yes

2. Is the protocol technically sound and planned in a manner that will lead to a meaningful outcome and allow testing the stated hypotheses?

Reviewer #2: Yes

3. Is the methodology feasible and described in sufficient detail to allow the work to be replicable?

Reviewer #2: Yes

4. Have the authors described where all data underlying the findings will be made available when the study is complete?

Reviewer #2: Yes

5. Is the manuscript presented in an intelligible fashion and written in standard English?

Reviewer #2: Yes

6. Review Comments to the Author

You may also provide optional suggestions and comments to authors that they might find helpful in planning their study.

Reviewer #2: I thank the authors who have made a great effort to address all the aspects of my comments. The paper is ready to be published.

7. PLOS authors have the option to publish the peer review history of their article (what does this mean?). If published, this will include your full peer review and any attached files.

Reviewer #2: No

---

## [Editor Report · Acceptance letter]

29 May 2024

PONE-D-23-35302R1 

PLOS ONE

Dear Dr. Njuma Libwea, 

I'm pleased to inform you that your manuscript has been deemed suitable for publication in PLOS ONE. Congratulations! Your manuscript is now being handed over to our production team.

Kind regards, 

on behalf of

Dr. Dong Keon Yon 

Academic Editor

PLOS ONE